# RelayGS: Reconstructing High-Fidelity Dynamic Scenes with Large-Scale and Complex Motions via Relay Gaussians

## Abstract

Reconstructing dynamic scenes with large-scale and complex motions—such as those in sports events—remains a significant challenge. Recent techniques like Neural Radiance Field and Gaussian Splatting have shown promise but often struggle with scenes involving substantial movement. In this paper, we propose **RelayGS**, a novel dynamic scene reconstruction method based on Gaussian Splatting, specifically designed to represent and learn large-scale complex motion patterns in highly dynamic scenes. Our RelayGS consists of three key stages. First, we learn the fundamental scene structure from all frames without considering temporal information and employ a learnable mask to decouple the highly dynamic foreground from the background exhibiting minimal motion. Second, we partition the scene into temporal segments, each consisting of several consecutive multi-view frames. For each segment, we replicate the foreground Gaussians, dubbed **Relay Gaussians**, as they are designed to act as relay nodes along the large-scale motion trajectory. By creating pseudo-views from frames uniformly selected from the segment, we optimize and densify foreground Relay Gaussians, further simplify and decompose large-scale motion trajectories into smaller, more manageable segments. Finally, we leverage HexPlane and lightweight MLPs to jointly learn the scene's temporal motion field and refine the canonical Gaussians. We conduct extensive experiments on two dynamic scene datasets featuring large and complex motions to demonstrate the effectiveness of our RelayGS. RelayGS outperforms state-of-the-arts by more than 1 dB in PSNR, and successfully reconstructs real-world basketball game scenes in a much more complete and coherent manner, whereas previous methods usually struggle to capture the complex motion of players.

## 1 Introduction

Dynamic scene reconstruction plays a pivotal role in a wide range of applications that demand immersive and interactive environments, including virtual reality, metaverse, and free-viewpoint videos. However, achieving high-fidelity reconstruction of dynamic scenes with large-scale and complex motions from multi-view videos remains a substantial challenge.

The recently emerged Gaussian Splatting (3DGS) Kerbl et al. (2023) has significantly advanced 3D reconstruction, inspiring numerous methods that enhance both reconstruction efficiency and quality. Compared to its predecessor, Neural Radiance Field (NeRF) Mildenhall et al. (2020), 3DGS uses Gaussian ellipsoids as primitives to explicitly represent 3D scenes, enabling real-time $1080p$ rendering via a rasterized pipeline. Similar to dynamic NeRF methods Pumarola et al. (2021); Park et al. (2021a;b), 3DGS has also been extended to dynamic scene reconstruction Yang et al. (2024a;b); Liu et al. (2024); Huang et al. (2024); Lu et al. (2024); Mihajlovic et al. (2024); Diwen Wan (2024), typically employing a framework that combines canonical space representations with implicit motion fields learned via neural networks. While work well for small-scale motions in public datasets, these methods encounter difficulties when handling large-scale and complex motions in real-world scenarios. For instance, in dynamic settings like basketball games, where multiple players move rapidly across the court, existing methods struggle to accurately capture the fast and large-scale movements of these players. This limitation arises from the coupling of canonical Gaussian representation learning with implicit neural motion field learning, which complicates optimization. Neural networks not only

find it challenging to predict large motions but also tend to overfit the dominant small motions in the scene, limiting their ability to model extensive complex movements.

We believe that one crucial aspect in addressing the challenge of the aforementioned problem is decoupling the highly dynamic foreground from the background with minimal motion. By isolating the foreground, we can better capture the large and complex motion trajectories of moving objects, while minimizing interference from the background. Moreover, MLPs, as a classical solution for representing motion fields, can efficiently handle the dynamic of dominant background content. The primary challenge, however, lies in modeling large, non-rigid, and complex foreground motions, which can be addressed by decomposing these motion trajectories into shorter, simpler segments.

In this paper, we propose **RelayGS**, a novel method for reconstructing dynamic scenes with large-scale, complex motions, consisting of the following three key stages:

• **I)** We learn a static initial scene from all frames ignoring temporal information. However, this can only capture the shared background of the entire scene. To address this, we introduce a *learnable mask* to distinguish whether a Gaussian belongs to the foreground (high dynamics) or background (low dynamics). All Gaussians are used for rendering the first frame, while for other frames, only those with the mask equals 1 are used, enabling us to learn a coarse representation of both the shared background and initial foreground while effectively decoupling the two.

• **II)** We divide the scene along the timeline into segments, each containing several consecutive frames (*e.g.*, the *1st-16th* frames as one segment). For each segment, we copy the initial foreground Gaussians decoupled in the first stage, and warm it up as the current segment's foreground Gaussians, then uniformly select three frames (*e.g.*, frames *1, 8, 16*) within the segment, blending them to create pseudo-views that serve as ground truth views. These foreground Gaussians act as explicit intermediate points along the motion trajectory, which we refer to them as **Relay Gaussians**, breaks down the large, complex motion trajectories into smaller, more manageable motion segments.

• **III)** We utilize the HexPlane Cao & Johnson (2023) and lightweight MLPs to predict time-continuous implicit motion offsets from the explicit canonical Gaussians initialized from the previous stage. For the shared background Gaussians, we use one set of MLPs to predict temporal changes in Gaussian properties. For the foreground **Relay Gaussians** across all segments, we employ another set of MLPs and additionally introduce a learnable scaling factor for position changes, as they may require a larger range that cannot be fully captured by the MLP's predictions alone. This ensures that the foreground Relay Gaussians can more accurately reflect large and complex motions in the scene.

We conducted extensive experiments to validate the effectiveness of our **RelayGS**. On the publicly available PanopticSports dataset Joo et al. (2015), which features large-scale motions, our method outperforms the previous state-of-the-arts with **1 dB** improvement in PSNR. Moreover, on a more complex real-world VRU Basketball Games dataset VRU (2024), our method successfully reconstructs the scene in a much more complete and coherent manner, whereas previous methods usually struggled to capture the dynamic foreground content with complex motions. The contributions of this paper can be summarized as follows:

- We introduce a simple learnable mask that effectively decouples high dynamic foreground and low dynamic background Gaussians without relying on additional priors, while learning a more accurate and complete fundamental 3D Gaussian representation of the dynamic scene.

- We propose the temporal Relay Gaussians to decompose large-scale and complex motion trajectories into smaller, more manageable motion segments, simplifying the representation and learning of complex dynamics.

- We utilize distinct MLPs to predict motion changes for background Gaussians and foreground Relay Gaussians, along with a learnable scaling factor for the position changes of Relay Gaussians, enabling accurate capture of larger and more complex motions.

- We conduct extensive experiments on two real-world dynamic scene datasets featuring large-scale, complex motions, where our RelayGS significantly outperforms previous state-of-the-art methods, achieving a **1 dB** improvement in PSNR on PanopticSports dataset and delivering more complete and coherent reconstructions of complex, large-scale foreground motions.

## 2 RELATED WORK

**Dynamic Scene Representation.** **(1)** NeRF-based methods have advanced dynamic scene reconstruction using coordinate-based neural networks. D-NeRF Pumarola et al. (2021) introduced a deformation network that warps samples from a canonical space over time, enabling accurate dynamic scene representation. Extensions like Nerfies Park et al. (2021a) and HyperNeRF Park et al. (2021b) use per-frame deformation codes for flexible modeling without relying solely on temporal input. These methods aim to construct a deformation field that maps the canonical scene to dynamic frames, but often with high computational costs due to dense sampling. **(2)** In contrast, methods based on 3D Gaussian Splatting (3DGS) like 4D-GS Yang et al. (2024a) and D3DGS Yang et al. (2024b) employ a deformation network that processes Gaussian center positions and timestamps to model scene dynamics. Our work implements a 3DGS-based framework, benefiting from fast training, rendering, and explicit representation.

**Dynamic-Static Decoupling.** One of the challenge in dynamic scene reconstruction is separating foreground and background. **(1) Motion masks** simplify this process. S4D He et al. (2024) classifies Gaussian points through multi-view 2D masks and a Gaussian category voting algorithm, effectively separating dynamic objects and static backgrounds. Similarly, EgoGaussian Zhang et al. (2024), and SC-4DGS Li et al. (2024a) also utilize pre-trained segmentation models to obtain motion masks. The limitations of these methods lie in their reliance on 2D masks and their tendency to focus only on areas with significant motion regions. **(2)** Some methods Guo et al. (2024); Liang et al. (2023) adopt the solution of lifting 2D **optical flow** to 3D. Katsumata et al. (2024) align Gaussian motion with optical flow data, improving spatiotemporal consistency, while GauFRe Liang et al. (2023) achieves the separation of static and dynamic elements based on optical flow-based motion detection. However, these approaches rely on pre-trained priors for optical flow, depth, or tracking. Our method bypasses motion priors, using a learnable mask to decouple dynamic foreground from relatively static background, making it more adaptable to complex scenarios and motion patterns.

**Dynamic Modeling.** Another key point is how to model the spatiotemporal dynamics. **(1)** The most intuitive approach Yang et al. (2024a); Duan et al. (2024) predicts temporal changes in position or appearance attributes using a **deformation field**. For instance, Gaussian-Flow Lin et al. (2024) combines polynomial fitting in the time domain and Fourier series fitting in the frequency domain to deform 3D Gaussian attributes over time. However, these methods are limited by model capacity and struggle with long-term motion. **(2)** To address motion complexity, some works Wu et al. (2024); Sun et al. (2024) employ **latent embeddings** to implicitly model the dynamics. E-D3DGS Bae et al. (2024) and Li et al. (2024b) assign embeddings to each Gaussian, predicting changes over time through MLPs, while DynMF Kratimenos et al. (2024) learns latent trajectories for Gaussian groups. However, this implicit embedding still struggles to maintain stable and coherent modeling in large scenes and high dynamics. To mitigate this limitation, in this work, we combine implicit deformation fields with explicit trajectory initialization. By stacking multi-frame views to construct pseudo-supervision, we create reasonable motion trajectories, reducing the deformation field's burden and enabling stable modeling of large-scale, dynamic, and long-term scenes.

## 3 PRELIMINARIES

**3D Gaussian Splatting.** 3D Gaussian Splatting Kerbl et al. (2023) explicitly represents scenes using anisotropic 3D Gaussian primitives, mathematically formulated as:

$$G(\mathbf{x}) = e^{-\frac{1}{2}(\mathbf{x}-\boldsymbol{\mu})^T \boldsymbol{\Sigma}^{-1}(\mathbf{x}-\boldsymbol{\mu})}, \quad \boldsymbol{\Sigma} = \mathbf{R}\mathbf{S}\mathbf{S}^T\mathbf{R}^T, \tag{1}$$

where the mean vector $\boldsymbol{\mu}$ and covariance matrix $\boldsymbol{\Sigma}$ respectively characterize the central position and geometric shape. The matrix $\boldsymbol{\Sigma}$ is decomposed into a scaling matrix $\mathbf{S} = \mathrm{diag}(s_x, s_y, s_z) \in \mathbb{R}^3$ and a rotation matrix $\mathbf{R} \in SO(3)$ to ensure physical meaning and facilitate optimization.

Rendering is performed by blending the contributions of $N$ overlapping Gaussian primitives at each pixel, taking into account their depth-ordering to ensure correct compositing, expressed as:

$$C = \sum_{i \in N} \mathbf{c}_i \alpha_i \prod_{j=1}^{i-1}(1 - \alpha_j), \tag{2}$$

where $\mathbf{c}_i$, $\alpha_i$ represents the color and blending weight of the $i^{th}$ Gaussian, respectively.

The training of 3D Gaussian Splatting alternates between parameter optimization and density control. Parameter optimization is supervised by the $\mathcal{L}_1$ loss and D-SSIM term:

$$\mathcal{L} = (1 - \lambda)\mathcal{L}_1 + \lambda\mathcal{L}_{\text{D-SSIM}} \tag{3}$$

where $\lambda$ is typically set to 0.2. Meanwhile, density control manages Gaussian cloning and splitting to address over-reconstruction and under-reconstruction.

**4D Gaussian Splatting.** 4D-GS Wu et al. (2024) builds upon the 3DGS by adding a deformation field, which consists of a Spatial-Temporal Structure Encoder $\mathcal{H}$ and a Multi-head Gaussian Deformation Decoder $\mathcal{D}$. The deformation field will cause the 3D Gaussians to undergo position shift, scaling, and rotation over time.

The position of 3D Gaussian $\boldsymbol{\mu}$ and time $t$ are input into the Spatial-Temporal Structure Encoder $\mathcal{H}$ together. The encoder, including a multi-resolution HexPlane $R_l(i, j)$ and a lightweight MLP $\phi_d$, fuses temporal and spatial information to obtain features $f_d$. Specifically, the mean value of 3D Gaussians $\boldsymbol{\mu} = (x, y, z)$ and time $t$ are combined in pairs to generate six multi-resolution planes, which is defined by $\{R_l(i, j)|(i, j) \in \{(x, y), (x, z), (y, z), (x, t), (y, t), (z, t)\}, l \in \{1, 2\}\}$, where $l$ is upsampling scale. Each voxel module will output the feature of neural voxels $f_h \in \mathbb{R}^{h*l}$ through bilinear interpolation for querying the voxel features, where $h$ is the hidden dim of features. Subsequently, the feature $f_h$ will be fused using a lightweight MLP $\phi_d$ by $f_d = \phi_d(f_h)$.

The output heads $\mathcal{D} = \{\phi_{\boldsymbol{\mu}}, \phi_r, \phi_s, \phi_o\}$ decode the features $f_d$ into predicted offsets for position $\Delta\boldsymbol{\mu} = \phi_{\boldsymbol{\mu}}(f_d)$, rotation $\Delta r = \phi_r(f_d)$, scaling $\Delta s = \phi_s(f_d)$ and opacity $\Delta\alpha = \phi_o(f_d)$, respectively. The deformed 3D Gaussian is expressed as $\mathcal{G}' = \{\boldsymbol{\mu} + \Delta\boldsymbol{\mu}, s + \Delta s, r + \Delta r, \alpha + \Delta\alpha, \mathcal{C}\}$, At time $t$, the 3D Gaussian $\mathcal{G}$ in the scene will be replaced by the deformed 3D Gaussian $\mathcal{G}'$ for rendering.

The optimization of 4D Gaussians is divided into two stages. The first stage is a warm-up period that uses only 3D Gaussians to optimize static scenes. In the second stage, the parameters of the HexPlane, MLPs, and 3D Gaussians are optimized simultaneously. The loss function comprises an $\mathcal{L}_1$ loss between the rendered image $\hat{I}$ and the GT image $I$ and a grid-based total variation loss $\mathcal{L}_{tv}$:

$$\mathcal{L} = |\hat{I} - I| + \mathcal{L}_{tv}. \tag{4}$$

## 4 METHODOLOGY

The proposed method, **RelayGS**, is designed to effectively tackle the challenge of reconstructing dynamic scenes with large-scale and complex motions by leveraging a combination of explicit and implicit representations. The method consists of three progressive stages, as shown in Fig. 1. In the **first stage**, we quickly learn an initial coarse representation of the scene without considering temporal information. This foundational stage allows us to capture the general structure of the scene and decouple the dynamic foreground, where large-scale motions may occur, from the relatively static background. In the **second stage**, we introduce **Relay Gaussians** to simplify and decompose large-scale motion trajectories into smaller, more manageable segments, allowing for a more efficient and detailed capture of dynamic content. In the **final stage**, we incorporate an implicit motion field through the use of HexPlane and lightweight MLPs. This stage refines the previously learned base Gaussian representations, enabling a full understanding of the scene's 4D spatiotemporal structure.

### 4.1 STAGE 1: INITIAL REPRESENTATION AND FOREGROUND-BACKGROUND DECOUPLING

The primary goal of this first stage is to construct the fundamental 3D structure of the dynamic scene. Previous method Wu et al. (2024) initialize a set of static Gaussians from sparse point clouds and jointly optimize them using all given frames without considering temporal information, *i.e.*, treating it as a static scene for initialization. This approach effectively captures the relatively static background of the scene, but struggles with the highly dynamic foreground.

The highly dynamic foreground, due to its significant positional variations across frames, cannot be easily initialized. For instance, even if some Gaussians can model dynamic foreground objects in a specific frame, due to the large motion of the objects, they may cause inconsistencies in another frame, resulting in large rendering errors. Under this initialization paradigm, the Gaussians representing such foreground objects would be noisy or automatically pruned.

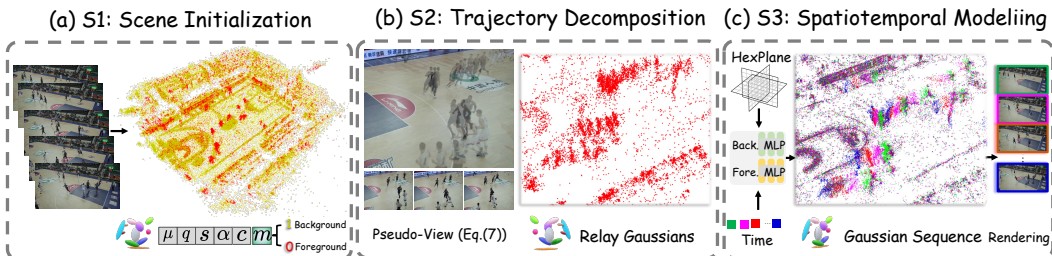

Figure 1: Framework of RelayGS. (a) First stage: Initialize the scene with all images and separate the relatively static background and dynamic foreground using a learnable mask (visualized as yellow and red). (b) Second stage: Construct pseudo-GT views through multi-view blending to generate Relay Gaaussians for decomposing long trajectories. (c) Third stage: Based on the HexPlane 4D representation, decode the foreground and background Gaussians using different MLPs to obtain time-dependent Gaussian sequences, and then render through the differentiable pipeline of 3DGS.

To address this limitation and learn the highly dynamic foreground simultaneously, we introduce a *"learnable mask"* for each Gaussian primitive to indicate whether it belongs to the highly dynamic foreground or the relatively static background. This idea is inspired by the Compact3DGS Lee et al. (2024), which was originally used to assess the importance of each Gaussian primitive in static scenes for rendering quality, allowing for pruning and compression, thereby reducing storage overhead while maintaining rendering quality. The formulation is written as:

$$\mathbf{M}_n = \mathrm{sg}(\mathbb{1}[\sigma(\mathbf{m}_n) > \epsilon] - \sigma(\mathbf{m}_n)) + \sigma(\mathbf{m}_n), \tag{5}$$

$$\hat{\boldsymbol{\alpha}}_n = \mathbf{M}_n \boldsymbol{\alpha}_n, \tag{6}$$

where $n$ is the index among all $N$ Gaussians, $\epsilon$ is the masking threshold, $\mathbf{m} \in \mathbb{R}^N$ is the learnable mask parameter, $\mathbf{M} \in \{0,1\}^N$ is the generated binary masks, $\mathrm{sg}(\cdot)$ is the stop gradient operator, and $\mathbb{1}[\cdot]$ and $\sigma(\cdot)$ are indicator and sigmoid function, respectively. The $\boldsymbol{\alpha}_n$ and $\hat{\boldsymbol{\alpha}}_n$ are the opacity before and after applying the mask, respectively.

We use all Gaussians to render the views for the first frame. However, for other frames, only the Gaussians where $\mathbf{M}_n$ equals 1 are used for rendering, which is implemented by Eq. (6). In this way, we can effectively decouple the base Gaussians into two groups, as shown in Fig. 1(a), allowing the separation of the highly dynamic foreground from the background with minimal motion.

This initialization process not only allows us to learn a better foundational scene representation compared to previous methods, but the decoupling of the foreground and background also plays a significant role in subsequent stages, as detailed in the following sections.

### 4.2 STAGE 2: LARGE MOTION TRAJECTORY DECOMPOSITION BY RELAY GAUSSIANS

**Segments along timeline.** The foreground objects in highly dynamic scenes often undergo significant movements across frames, making it difficult to fully capture their large-scale motion trajectory with a single set of canonical Gaussians. To address this issue, in the second stage, we aim to explicitly decompose the large motion trajectory of the dynamic foreground into smaller, more manageable segments. In our implementation, consecutive $k{=}16$ frames are treated as one segment, *i.e.*, the $1st$-$16th$ frames form the first segment, followed by subsequent segments.

**Relay Gaussians.** Since motion trajectories are continuous over time, this segmentation also effectively breaks down the large motion trajectory into smaller segments, each representing a portion of the overall motion trajectory. For each segment, we replicate the dynamic foreground Gaussians from the first stage and distinguish them as **Relay Gaussians**, as they are designed to act as relay nodes along the large-scale motion trajectory, passing on critical information about the object's position and movement across different time intervals.

**Pseudo-Views.** For each segment, we construct pseudo-views by blending $p = 3$ uniformly selected frames (*e.g.*, frames 1, 8, and 16 in the first segment) for supervision. Let the three selected frames in a segment be denoted as $I_{t_1}, I_{t_2}, I_{t_3}$. The pseudo-view $I_{\mathrm{pseudo}}$ for this segment is then constructed as:

$$I_{\mathrm{pseudo}} = \beta_1 I_{t_1} + \beta_2 I_{t_2} + \beta_3 I_{t_3}, \tag{7}$$

where $\beta_1 + \beta_2 + \beta_3 = 1$ are blending weights applied to the selected frames, typically chosen based on frame importance or uniform blending. In this work, we use the strightforward uniform blending, *i.e.*, $\beta_1 = \beta_2 = \beta_3 = \frac{1}{3}$, for conciseness. $I_{\text{pseudo}}$ replaces the $I$ in Eq. (4) for optimization. These pseudo-views capture snapshots of the foreground at different time steps, as shown in Fig. 1 (b), providing a richer representation for optimizing the Relay Gaussians, ensuring they more accurately capture the motion trajectory within each segment.

By leveraging Relay Gaussians to decompose large-scale motion trajectories into smaller, more manageable segments, we reduce the complexity of handling dynamic motions, which will become evident in the final learning stage.

### 4.3 STAGE 3: 4D SPATIOTEMPORAL MODELIING AND OPTIMIZATION

**4D representation.** To achieve a complete 4D dynamic scene representation, it is crucial to incorporate temporal information, typically through an implicit motion field. In this work, we adopt the representative **4D-GS** framework. This choice is driven by the efficiency of HexPlane and MLPs in encoding spatiotemporal data and their flexibility in modeling dynamic motion. Additionally, the simplicity of the HexPlane-MLP combination allows for scalable optimization. It is worth noting that our RelayGS is ***flexible and can be extended*** to leverage other motion fields, such as Per-GS Bae et al. (2024), to further enhance motion representation, which will be explored in future works.

**Foreground-background isolation.** To avoid overfitting to small motions due to all Gaussians sharing MLPs, we propose a divide-and-conquer strategy. For the background Gaussians, we utilize a dedicated set of MLPs that predict the temporal changes in their positions and other attributes relative to their base Gaussians. For the foreground Relay Gaussians, another set of MLPs models their time-varying positions and attributes throughout the motion trajectory, as shown in Fig. 1 (c).

**Position deformation scaling.** To further enhance the nonlinear capability of the model to better learn more complex motion patterns, for each Relay Gaussian, we introduce a learnable scaling factor $\gamma \in \mathbb{R}^3$ that accounts for larger motion ranges, which may not be fully captured by the MLP alone. This factor ensures that the Relay Gaussians can adapt to complex motions that extend beyond the capacity of standard MLP predictions.

$$\boldsymbol{\mu} \leftarrow \boldsymbol{\mu} + (1 + e^{\boldsymbol{\gamma}}) \cdot \Delta\boldsymbol{\mu}. \tag{8}$$

Through this stage, we achieve a comprehensive 4D scene reconstruction, integrating both spatial and temporal dynamics. The optimization performed here refines the learned Gaussians and finalizes the motion trajectories, resulting in a coherent and accurate representation of the entire dynamic scene.

## 5 EXPERIMENT

### 5.1 EXPERIMENTAL SETUP

In this work, we primarily focus on addressing large-scale and complex motion in dynamic scenes. To evaluate our RelayGS's effectiveness, we conduct experiments on two representative datasets:
**PanopticSports Dataset.** This is a subset of the CMU Panoptic Studio dataset Joo et al. (2015), containing 6 dynamic sports scenes: Juggle, Box, Softball, Tennis, Football and Basketball. Each scene has a resolution of $640 \times 360$ and spans 150 frames, captured at 30 FPS. The data was collected using 31 static cameras, of which 27 are used for training and 4 for testing (cameras 0, 10, 15, and 30).
**VRU Basketball Games Dataset.** This dataset VRU (2024) contains two real-world basketball game scenes, "GZ" and "DG4". Each was captured in an indoor basketball court using 34 fixed, synchronized cameras, evenly distributed around the court to cover 360 degrees. The sequences span 10 seconds, with a resolution of $1920 \times 1080$ at 25 FPS, resulting in 250 frames per sequence. Of the 34 cameras, 30 are used for training, while 4 (cameras 0, 10, 20, and 30) are reserved for testing. More details of these datasets can be found in the Appendix.

**Implementation**. Our implementation is based on the open-source 4D-GS Wu et al. (2024) code. In the first stage, 3D Gaussians are initialize using sparse point cloud, following the 3DGS Kerbl et al. (2023) and 4D-GS, and a mask attribute is assigned to each Gaussian, initialized to 2, which results

Table 1: Quantitative results on the VRU Basketball Games dataset. "ST-GS[†]" uses point clouds of uniformly selected 16 frames for initialization to ensure a fair comparison with our method, while "ST-GS" utilizes point clouds of all 250 frames, the default setting for their method.

| Method | GZ | | | | DG4 | | | |
|---|---|---|---|---|---|---|---|---|
| | PSNR (dB ↑) | Storage (MB ↓) | Train (mins ↓) | Render (fps ↑) | PSNR (dB ↑) | Storage (MB ↓) | Train (mins ↓) | Render (fps ↑) |
| 4D-GS | 25.83 | 42 | 63 | 88 | 25.17 | 45 | 62 | 80 |
| ST-GS | 27.32 | 400 | 107 | 143 | 26.79 | 360 | 112 | 134 |
| ST-GS[†] | 26.49 | 35 | 64 | 264 | 25.79 | 40 | 64 | 236 |
| E-D3DGS | 26.14 | 113 | 224 | 35 | 25.06 | 136 | 301 | 27 |
| **RelayGS (Ours)** | 28.06 | 200 | 105 | 74 | 26.94 | 191 | 107 | 69 |

Table 2: Quantitative results on the PanopticSports dataset. "Dynamic3DGS" and "D-MiSo" data are partially taken directly from their original papers or estimated based on the paper and available code.

| Method | Juggle | | | Boxes | | | Softball | | |
|---|---|---|---|---|---|---|---|---|---|
| | PSNR (dB ↑) | Storage (MB ↓) | Train (mins ↓) | PSNR (dB ↑) | Storage (MB ↓) | Train (mins ↓) | PSNR (dB ↑) | Storage (MB ↓) | Train (mins ↓) |
| Dynamic3DGS | 29.48 | 221 | 107 | 29.46 | 221 | 108 | 28.43 | 221 | 116 |
| 4D-GS | 28.19 | 48 | 30 | 27.67 | 47 | 29 | 27.41 | 46 | 29 |
| E-D3DGS | 26.54 | 36 | 95 | 26.78 | 33 | 100 | 26.01 | 33 | 80 |
| D-MiSo | 29.79 | - | - | 29.39 | - | - | 28.60 | - | - |
| **RelayGS (Ours)** | 30.06 | 31 | 48 | 29.99 | 30 | 48 | 30.20 | 33 | 48 |
| | Tennis | | | Football | | | Basketball | | |
| | PSNR (dB ↑) | Storage (MB ↓) | Train (mins ↓) | PSNR (dB ↑) | Storage (MB ↓) | Train (mins ↓) | PSNR (dB ↑) | Storage (MB ↓) | Train (mins ↓) |
| Dynamic3DGS | 28.11 | 221 | 101 | 28.49 | 221 | 114 | 28.22 | 221 | 113 |
| 4D-GS | 27.49 | 45 | 29 | 26.67 | 54 | 33 | 27.72 | 37 | 24 |
| E-D3DGS | 27.41 | 31 | 74 | 25.93 | 33 | 76 | 26.48 | 35 | 87 |
| D-MiSo | 29.02 | - | - | 28.99 | - | - | 28.49 | - | - |
| **RelayGS (Ours)** | 30.21 | 31 | 48 | 30.23 | 37 | 48 | 29.77 | 51 | 48 |

in a value close to 1 after sigmoid activation. The optimization running for 3,000 steps with periodic densification. Then, the Gaussians are separated into foreground and background based on the learned mask values. In the second stage, the scene is divided into segments, each consisting of $k=16$ frames in our experiments. This stage is trained for 14,000 steps. In the third stage, we initialize HexPlane and MLPs in the same manner as 4D-GS. However, we configure two separate sets of MLPs: one for background Gaussians and the other for Relay Gaussians. Both sets are responsible for predicting the changes in the four Gaussian attributes—position, scaling, rotation, and opacity—over time. We do not include the spherical harmonics MLP, as it increases the model size and reduces rendering speed without providing notable performance gains. Additionally, the $\gamma$ is initialized to 0. This stage is trained for 20,000 steps. For the PanopticSports dataset, multi-view color inconsistencies are present, so we apply a learnable channel-wise affine color tune for each camera, following Dynamic3DGS. For VRU scenes, we optimize using $2\times$ downsampled views to reduce time cost. All experiments were conducted on an NVIDIA RTX 4090 GPU with batch size 4. The learning rate and densification settings are consistent across all three stages, more details can be found in the Appendix.

## 5.2 EXPERIMENTAL RESULTS

**Quantitative Comparison.** We compare our RelayGS with several state-of-the-art methods, including 4D-GS Wu et al. (2024), Dynamic3DGS Luiten et al. (2024), ST-GS Li et al. (2024b),

| GT | Ours | ST-GS | 4D-GS |
|---|---|---|---|

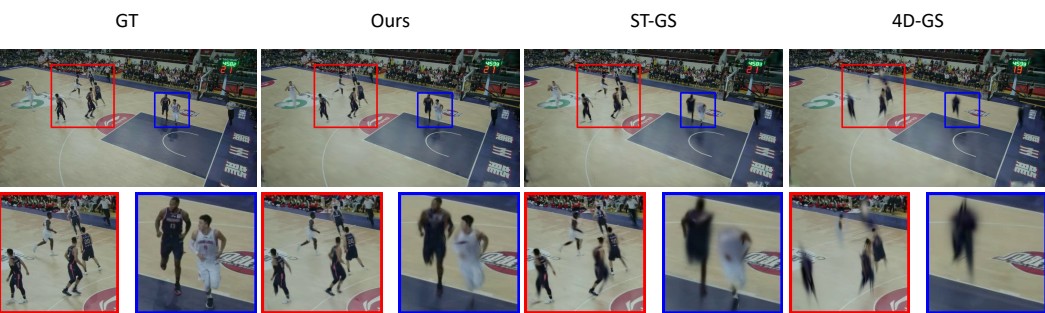

Figure 2: Qualitative comparisons on GZ scene of VRU Basketball Games dataset.

| GT | Ours | E-D3DGS | 4D-GS |
|---|---|---|---|

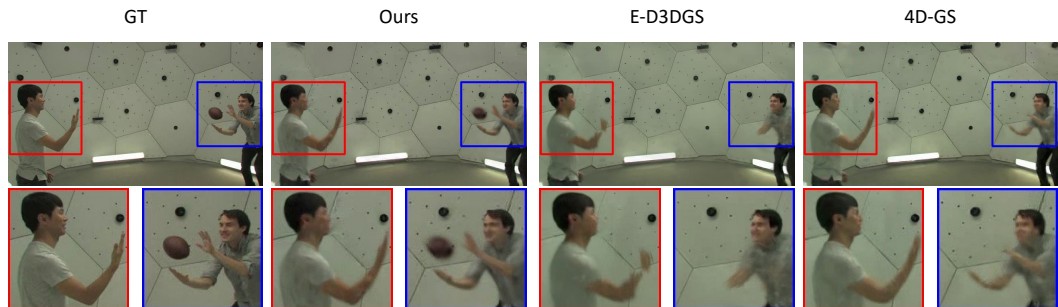

Figure 3: Qualitative comparisons on Football scene of PanopticSports dataset.

E-D3DGS Bae et al. (2024), and D-MiSo Waczyńska et al. (2024). The results are shown in Tab. 1 and Tab. 2. **(1) Quality**: Our RelayGS method consistently outperforms competitors in terms of reconstruction quality (*i.e.*, PSNR) on both datasets. Specifically, on the six scenes of the Panoptic-Sports dataset (see Tab. 2), RelayGS achieves PSNR improvements of 0.27 dB, 0.53 dB, 1.6 dB, 1.19 dB, 1.24 dB, and 1.28 dB, respectively, averaging a gain of 1.02 dB over the previous best methods. Compared to the baseline method 4D-GS, we achieve an average performance gain of 2.47 dB. On the more challenging VRU Basketball Games dataset (see Tab. 1), RelayGS outperforms the previous best method ST-GS and the baseline method 4D-GS by an average of 0.45 dB and 2 dB, respectively. It is worth ***noting*** that, although the PSNR difference compared with ST-GS appears small, the static floor occupies approximately 70% of the pixels in these VRU view images, meaning the quality improvement is more significant in the dynamic foreground regions. Additionally, ST-GS is heavily dependent on initialization, as it extracts sparse point clouds for each frame and then merges them as the initial scene. Since point clouds for each frame cannot be obtained in the PanopticSports dataset, ST-GS is not applicable. **(2) Efficiency**: While our method learns corresponding foreground content for each segment via Relay Gaussians, RelayGS strikes a good balance between reconstruction quality and efficiency factors such as storage, training time, and rendering speed compared to competitors, some of which achieve high storage efficiency but fall short in reconstruction quality. In contrast, our method demonstrates a clear advantage in storage efficiency, particularly on the PanopticSports dataset. Compared to the baseline method 4D-GS, RelayGS introduces an additional stage with Relay Gaussians, which increases the training time and slightly reduces the rendering speed in some tend. However, RelayGS still maintains a clear advantage in training time compared to other methods. While achieving high-quality reconstruction, we can also ensure a real-time rendering speed of around 70 fps on RTX 4090 GPU.

**Qualitative Analysis**  Fig. 2 and Fig. 3 show frames from two representative scenes with heavily featured foreground dynamic content. As seen, our RelayGS reconstructs the humans with greater clarity and completeness. This improvement is primarily due to the fact that, compared to our baseline, 4D-GS, our stage I not only learns the background Gaussians but also captures the foreground Gaussians. In our stage II, we further refine the foreground Gaussians by learning additional Gaussians that cover more of the motion trajectories, known as Relay Gaussians. ST-GS, although using point clouds from all 250 frames, obtains a denser sampling of motion trajectories. However, due to its simpler approach to modeling motion changes, it struggles to accurately capture the

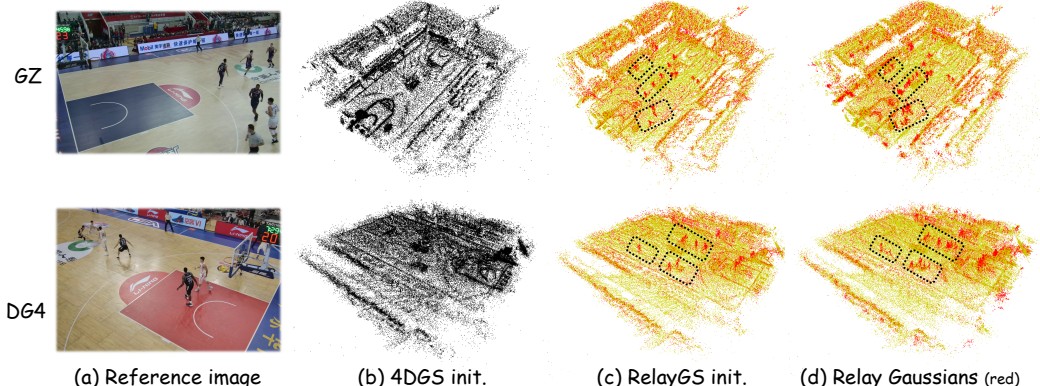

GZ

DG4

(a) Reference image     (b) 4DGS init.     (c) RelayGS init.     (d) Relay Gaussians (red)

Figure 4: The visualization of canonical 3D Gaussians. (a) Reference image of the scene. (b) Initialization by 4D-GS, with the dynamic Gaussian in the foreground almost eliminated. (c) Initialization by our method achieves separation of static background and dynamic foreground, visualized in different colors. (d) Relay Gaussians (red) generated in the second stage realize the decomposition of long trajectories.

Table 3: Ablation study on key design components. For detailed analysis, please refer to Sec. 5.3.

| Case | Method | | GZ | | Softball | |
|------|--------|--|------|--|----------|--|
| #1 | full method | | 28.06 | | 30.20 | |
| #2 | *w/o* | Segment along timeline | 26.07 | ↓1.99 | 29.42 | ↓0.78 |
| #3 | *w/o* | Stage II | 27.27 | ↓0.79 | 29.93 | ↓0.27 |
| #4 | *w/o* | Pseudo-Views | 27.80 | ↓0.26 | 30.00 | ↓0.20 |
| #5 | *w/o* | Fg-Bg Isolation | 27.80 | ↓0.26 | 30.07 | ↓0.13 |
| #6 | *w/o* | Scaling Factor $\gamma$ | 27.87 | ↓0.19 | 29.73 | ↓0.47 |

foreground with complex motions. This issue is more evident in the rendered videos, where ST-GS shows inconsistencies in the motion of the Gaussians associated with the same object, leading to flickering in the foreground. In contrast, our method, leveraging HexPlane encoding following 4D-GS, models temporally and spatially consistent motion, resulting in smoother and more coherent reconstructions. Additionally, both 4D-GS and E-D3DGS struggle to handle the large-scale motion of the ball in these scenes. In comparison, our method performs significantly better, although challenges remain. The relatively small and isolated ball with mostly empty space around it makes it difficult to track. Our second stage mitigates this issue to some extent by introducing Relay Gaussians, but it remains a challenging aspect due to the sparse Gaussians learned in the first stage. In summary, RelayGS not only achieves SOTA performance on quantitative metrics for the entire image but also demonstrates superior spatiotemporal modeling capabilities, particularly on foreground dynamic content. *We encourage readers to view the supplementary rendered videos for a more comprehensive understanding of our reconstruction results.*

**3D Gaussian visualization.** We visualize the canonical Gaussians learned at different stages, with the results shown in Fig. 4. As observed in Fig. 4 (b), in the baseline method 4D-GS, the canonical Gaussians learned in the first stage primarily represent the background, with very few Gaussians capturing the foreground. In contrast, in our method, the base Gaussians learned in the first stage include both background and foreground Gaussians, which can be distinguished by a binary mask, visualized in different colors in Fig. 4 (c). Furthermore, through the learning process in the second stage, our method is able to capture additional Relay Gaussians (red points in Fig. 4 (d)) along the motion trajectories of the foreground, significantly improving the representation of dynamic content.

Table 4: Ablation on number of frames per segment. The experiments are conducted on GZ scene of VRU Basketball Games dataset.

| $k$ | 8 | 16 | 32 | 64 | 128 |
|---|---|---|---|---|---|
| PSNR (dB) | 27.90 | 28.06 | 27.82 | 27.56 | 27.10 |

## 5.3 ABLATION STUDY

In Tab. 3, we present ablation studies on several key components of our method. The case #2 represents the configuration where no temporal segmentation is applied, and only a single global set of foreground Gaussians is used. This results in a significant performance drop, as it cannot effectively handle large-scale motion. In case #3, we remove the second stage of our method, directly replicating a set of foreground Gaussians for each segment and learning them jointly with the implicit motion field. This also leads to a notable performance decrease, especially in the more complex GZ scene. In case #4, we demonstrate the significance of multi-view synthesis pseudo-views, which enable the acquisition of richer Relay Gaussians representing trajectories. In cases #5 and #6, we conduct ablation studies on the setting of different MLPs for foreground-background isolation and the scaling factor $\gamma$ in the third stage, respectively. These results highlight the importance of our improvements for 4D spatiotemporal modeling.

In Tab. 4, we perform an ablation study on the length of each segment, *i.e.*, the number of frames included in each segment. As the segment length increases and the number of segments decreases, the motion trajectory within each segment becomes larger, leading to a gradual decline in performance. However, choosing the $k$ value too small will increase the training cost and not result in a significant performance improvement. Based on experience, we set $k=16$ as the default selection.

## 6 CONCLUSION

This paper proposes RelayGS, a novel method specifically designed to address the challenges of reconstructing dynamic scenes with large-scale and complex motions. We first learn the basic structure of the scene and, through a learnable mask, simultaneously capture the shared background and the foreground of the initial frame, achieving effective decoupling of dynamic foreground and relatively static background Gaussians. Then, we divide the scene into segments along the temporal dimension, replicating and learning a set of foreground Gaussians for each segment. The training views are constructed using pseudo-views by blending three frames within the segment. These foreground Gaussians are referred to as Relay Gaussians, which decompose the complex, large-scale motion trajectories into smaller, manageable segments. Finally, we further optimize the spatiotemporal representation of both the background Gaussians and foreground Relay Gaussians. Extensive experiments demonstrate that RelayGS outperforms state-of-the-art methods on two real-world datasets with large-scale motions, achieving significant improvements in reconstruction quality. Additionally, our method strikes a balance between reconstruction quality and storage efficiency, making it well-suited for real-world applications involving complex motions.

**Limitation.** While our method achieves significant performance advantages, it still faces some challenges. **(1)** Insufficient motion modeling of small but fast-moving objects. This is due to the limited pixel coverage of these objects, insufficient camera view coverage, and sparse base surrounding Gaussians, which make it difficult to accurately capture and reconstruct their motion. **(2)** Our method for segmenting the scene and constructing pseudo-views is relatively straightforward. In practice, the segmentation should adapt to the complexity of the motion in the scene, allowing for more precise divisions. Additionally, rather than uniformly selecting frames, a more adaptive approach would involve selecting frames along the motion trajectory in a way that better captures the motion dynamics. This would lead to learning more optimal Relay Gaussians, ultimately improving the accuracy of motion representation.

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

APPENDIX

This appendix provides additional material to supplement the main text.

## A  DATASET DETAILS

**PanopticSports Dataset.** The cameras are temporally aligned with accurate intrinsic and extrinsic parameters. Positioned in a roughly hemispherical arrangement around the area of interest in the middle of the capture studio, the cameras provide comprehensive coverage of the scene. The images are undistorted using the provided distortion parameters and resized to 640 × 360. The dataset provides a point cloud generated by 10 available depth cameras for each scene. In our experiments, this point cloud is first downsampled to approximately 35,000 points, which are then used to initialize the Gaussian primitives. Each scene involves one or two moving persons and some moving objects, while the background remains completely static. Additionally, the foreground colors are quite similar to the background, which further increases the difficulty of scene reconstruction due to the reduced contrast between the foreground and background elements.

**VRU Basketball Games Dataset.** The camera poses and distortion parameters were estimated using the first frame from all 34 views by COLMAP Schonberger & Frahm (2016), and all frames were undistorted accordingly. After undistortion, the resolution slightly increases, and we did not resize the images back to 1920×1080. Following the 4D-GS Wu et al. (2024) method, a point cloud was generated and downsampled to approximately 80,000 points for initializing the Gaussian primitives. Each scene includes multiple basketball players, a basketball, scoreboards, advertisement banners, and thousands of spectators. The basketball players and the basketball exhibit fast and large-scale movements with highly complex motion patterns, including non-rigid deformations. The scoreboards and banners also dynamically change over time, and even the background spectators are not completely static, as some exhibit subtle movements. Additionally, the physical scale of the scene is significantly larger than previously available dynamic scene datasets, making it highly challenging to reconstruct.

## B  MORE IMPLEMENTATION DETAILS

Our method employs slightly different settings for learning rates and densification thresholds between the foreground and background Gaussians. The background learning rates are similar to those used in previous methods, with the initial learning rate for position set to 2e-4 and the minimum learning rate to 1e-5. For the foreground Gaussians, the initial learning rate for position is set to 1e-3. The gradient threshold for densification is 1e-4, which is half of the threshold used for the background. Additionally, the scaling threshold for densification is set to 1e-3 for the foreground, which is 0.1 times that of the background. These settings encourage the foreground Gaussians to be smaller and split faster than the background Gaussians. More detailed experimental settings will be released in our future open-source code to better support reproducible research.

## C  ADDITIONAL QUALITY COMPARISON RESULTS

We present the quality comparison on other scenes from the two datasets in Figures 5 to 10. The visual results clearly demonstrate that our method consistently achieves significantly better visual quality compared to competitive counterparts across different scenes from both datasets, proving the generalization ability of our RelayGS approach.

## D  ADDITIONAL EXPERIMENTAL RESULTS

The goal of the first two stages of our method is to learn a more robust base Gaussian representation, simplifying complex motion patterns in the scene and preparing for full learning in the final stage. Using low-resolution views during these stages produces comparable results while significantly reducing training time. Additionally, we observed that our method performs more effectively at low resolutions, resulting in a larger performance gap compared to counterpart methods. The results are presented in Tab. 5, further reinforcing the superiority of our approach in motion learning.

Table 5: Quantitative results on the VRU Basketball Games dataset at half resolution. "ST-GS" utilizes point clouds of all 250 frames, the default setting for their method.

| Method | PSNR (dB ↑) | |
|---|---|---|
| | GZ | DG4 |
| ST-GS | 27.61 | 26.87 |
| E-D3DGS | 26.33 | 25.39 |
| **RelayGS (Ours)** | 28.97 | 27.50 |

In Fig. 11, we provide additional visualization results of Relay Gaussians on the PanopticSports dataset, showcasing how our method learns Relay Gaussians for large-scale dynamic content.

In the Supplementary Material, we provide a zip file that contains 3 videos: `VRU_GZ_GT.mp4`, `VRU_GZ_RelayGS_PSNR-28.06.mp4`, and `VRU_GZ_ST-GS_PSNR-27.32.mp4`. These videos represent, respectively, the ground truth videos from four test views, the videos rendered from our RelayGS method, and the videos rendered from the ST-GS Li et al. (2024b) method initialized with the sparse point clouds of all 250 frames. From these videos, the superior reconstruction quality and motion coherence of our method compared to ST-GS can be clearly observed.

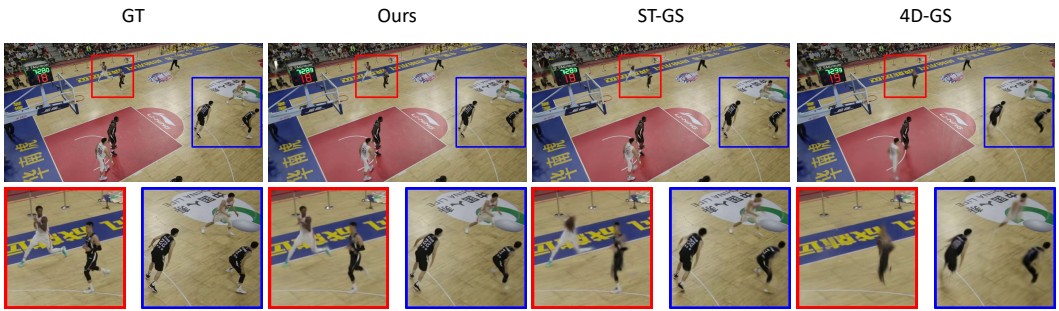

Figure 5: Qualitative comparisons on DG4 scene of VRU Basketball Games dataset.

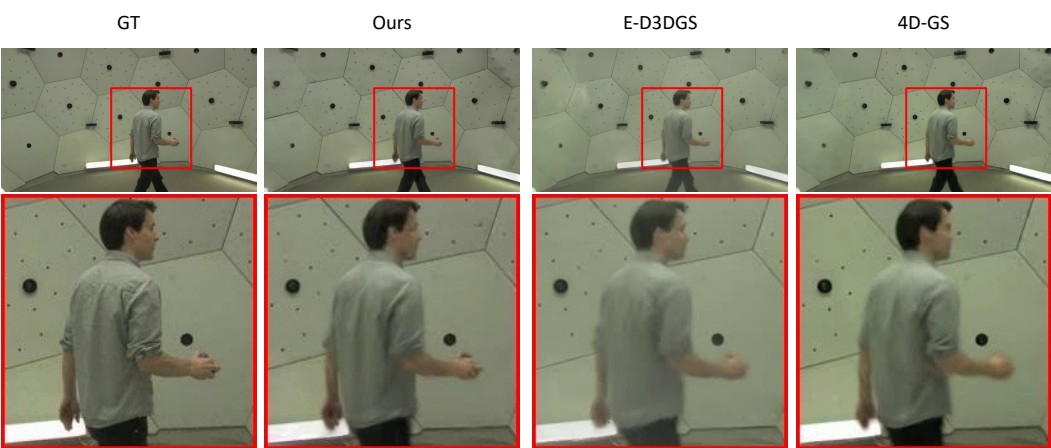

Figure 6: Qualitative comparisons on Juggle scene of PanopticSports dataset.

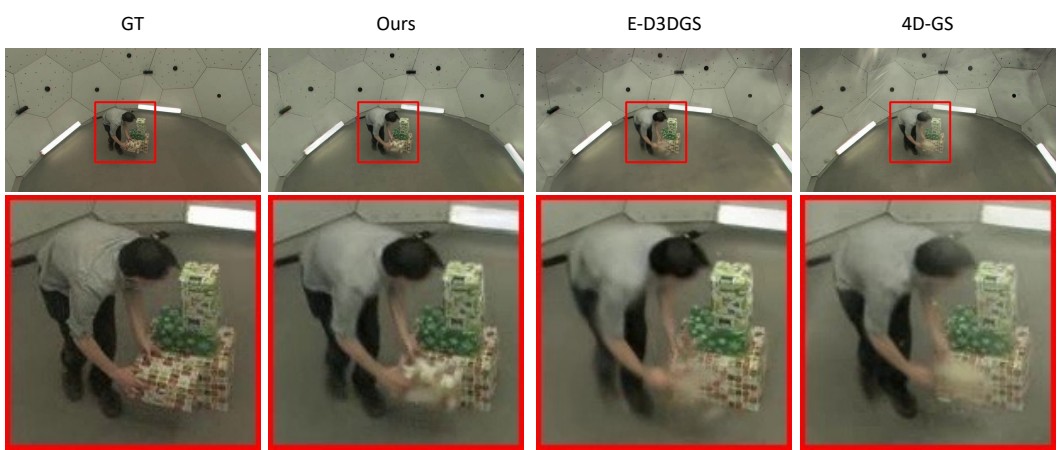

Figure 7: Qualitative comparisons on Boxes scene of PanopticSports dataset.

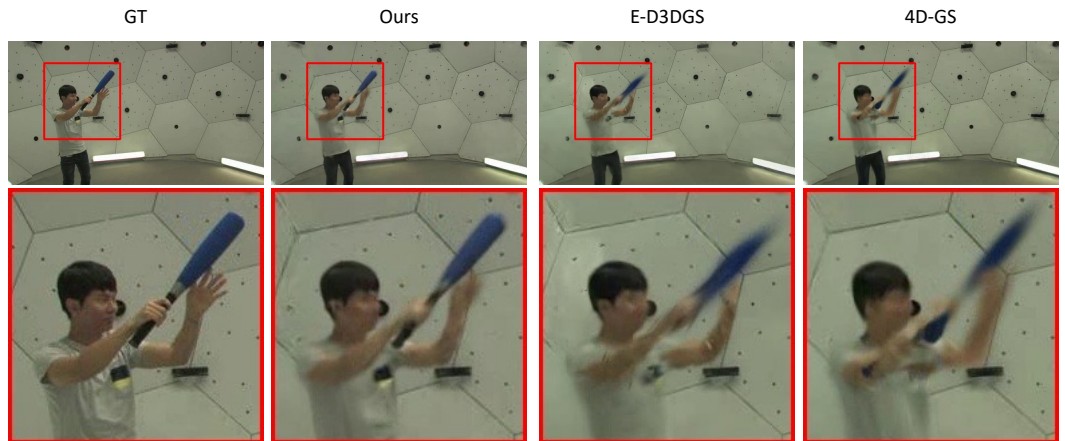

Figure 8: Qualitative comparisons on Softball scene of PanopticSports dataset.

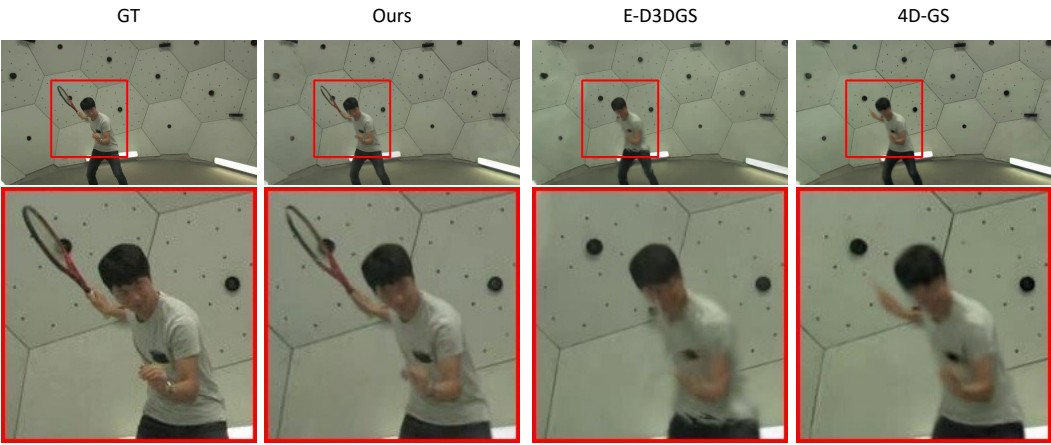

Figure 9: Qualitative comparisons on Tennis scene of PanopticSports dataset.

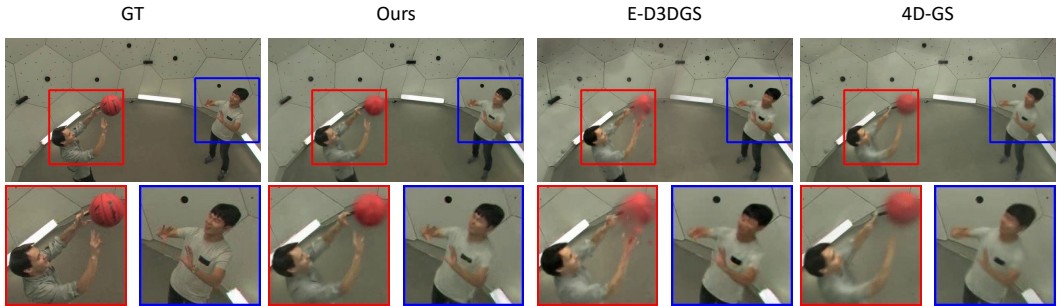

Figure 10: Qualitative comparisons on Basketball scene of PanopticSports dataset.

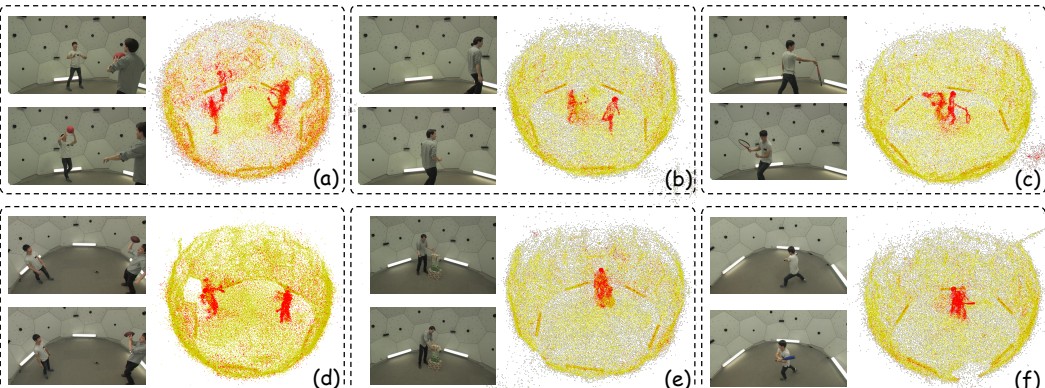

Figure 11: Visualizations of the second-stage dynamic foreground Relay Gaussians (red points) in 6 scenes of the PanopticSports dataset. (a)-(c) show people in the foreground with larger motion amplitudes, generating more dispersed trajectories. (d)-(f) show people in the foreground with smaller motion amplitudes, generating more concentrated trajectories.

