# OpenReview forum: "RelayGS: Reconstructing High-Fidelity Dynamic Scenes with Large-Scale and Complex Motions via Relay Gaussians"
_ICLR.cc/2025/Conference — ICLR 2025 Conference Withdrawn Submission_

### Official Review · Reviewer_uUUg · 2024-10-17

**Soundness:** 3
**Presentation:** 2
**Contribution:** 2
**Rating:** 5
**Confidence:** 4

**Summary:**

The paper presents a multi-stage dynamic 3D GS pipeline for large-scale scenes with multiple independent, high-frequency motions. In the first stage, the static (low-motion) background is initialized. The second stage initializes relay GSs for large-motion foreground objects. In the final stage, both the foreground and background are jointly optimized across the temporal dimension using deformation fields. The proposed method is evaluated on a novel basketball dataset and a public sports dataset, achieving state-of-the-art (SOTA) results in both cases.

**Strengths:**

1. The defined task is compelling. I agree with the author that current methods do not effectively handle large scenes with multiple dynamic objects.
2. The design of the overall pipeline is solid. Decoupling the scene into parts with different motion patterns improves the reconstruction quality.
3. Based on the ablation study, the designed relay GS initialized by fused pesudo-views, along with temporal splitting improve the overall accuracy.
4. The purposed method achieves highest accuracy on two benchmarks.

**Weaknesses:**

1. I have concerns about the reproducibility of the method. Although the concepts of each stage are well described and make sense, important implementation details are missing. For instance, the feature dimension $h$ (line 178), the setup of the sets of MLPs (lines 289-293), and $ϵ$ (in eq 5) are not specified. Additionally, one aspect remains unclear clear to me. Each time interval segment has its own set of relay GS, which are initialized from the same GS set in stage 1 but optimized independently. I assume density control is applied as in the original 4DGS, which could alter the total number of GS. How are these fused in stage 3. Are they simply merged, or are they dynamically selected based on time?
2. The author mentioned that pseudo-views provide a richer representation than a single frame. However, there is no ablation study on the number of frames used to generate the pseudo-views. All experiments are conducted using a setup with 3 frames. Is the selection of the frame number also related to the length of the segment? For instance, in Table 4, does each experiment use the same number of frames (i.e., 3)?
3. Another main concern is the experimental setup. Two datasets are evaluated in total. On of which is a novel basketball dataset, which only contains two videos. Another sports dataset is not so widely used, which is also supported by the fact that Table 2 includes only 4 competing methods. While I full understand that the proposed method focuses on large scale scenes with multiple motions, it is more convincing if the author can provide the evaluation on more widely used dynamic dataset like Neural 3D Video[1]. Also, the competitors for the two datasets are not fully consistent. The second best method in Table 1 does not appear in Table 2, and vice versa. It has been partially demonstrated that segment the entire video into clips benefits the dynamic reconstruction [2]. So I believe providing the evaluation on more widely used dataset would be more convincing.
4. Some figures are difficult to interpret. For instance, the GS point cloud in Figure 1(c) and Figure 4 (c)&(d) are hard to read. In  Figure 4 (b)&(c), tiny difference can be observed, but their significance is unclear. From my perspective, (c) is actually better since the human body shape is more clear, but (d) generates better image. It is understandable that the author aims to show the reconstruction of the entire scene. However, I believe some zoomed-in visualizations, such as focusing on a single character over time would provide clearer insight.


[1] Li, Tianye, et al. "Neural 3d video synthesis from multi-view video." Proceedings of the IEEE/CVF Conference on Computer Vision and Pattern Recognition. 2022.

[2] Shaw, Richard, et al. "SWinGS: Sliding Windows for Dynamic 3D Gaussian Splatting."

**Questions:**

My questions are described in the Weaknesses section.

I am open to revise my rating if all my concerns are addressed.

---

### Official Review · Reviewer_9GMy · 2024-10-22

**Soundness:** 2
**Presentation:** 2
**Contribution:** 2
**Rating:** 3
**Confidence:** 4

**Summary:**

The paper considers the problem of fitting a dynamic 3D Gaussian mixture to a video captured from multiple static cameras to reconstruct it in 4D. Compared to prior works that looked at a similar problem, the paper:

* Introduces a new way of initially separating static and dynamic Gaussians via a learnable
* Proposes to break long videos into shorter segments that are reconstructed individually
* Combines various ideas such as using HexPlan to model motion (see 4D-GS) to capture the motion of the Gaussians

On six different multi-camera videos from two datasets, the method is shown to outperform competitors. This is particularly evident in the qualitative evaluation.

**Strengths:**

* The paper looks at a timely problem (4D reconstruction)
* Some design decisions are plausible and worth exploring, including processing the video into short segments, marking Gaussians as static or dynamic, etc
* The result are, empirically, good and qualitatively quite a lot better than prior works

**Weaknesses:**

The paper seems to be missing key information and I had a very hard time trying to fill in the blanks. I would be entirely unable to reproduce the method from the information that it is there. Unfortunately a very significant amount of rewriting is likely required before the details are clear enough for the paper to be accepted.

First some general observations:

1. The paper does not make all of its assumption clear from the beginning. For instance, I think it is implicitly assumed that the cameras are static, as otherwise it would not make much sense to combine frames linearly to obtain the pseudo-images (Eq. (7)). All datasets used in the experiments appear to have static cameras. However, such a crucial assumption is not explicitly stated.
2. The paper should better explain upfront what is the 3D Gaussian fitting/reconstruction strategy, even at the cost of briefly summarising how it is done in the prior works at the basis of this contribution. This to me was not clear until reading some experimental details in Section 5.1, and even then many details are obscure.
3. Because the cameras are static, if one discounts the dynamic part of the scene, then reconstructing from a single frame or all frames would result in near identical 3D GS reconstruction results. If these reconstructions are good, then, it means that the cameras are sufficient in number to instantaneously perform 3D GS reconstruction at every frame. Hence, in order to reconstruct the dynamic scene, we could simply reconstruct the scene from scratch at every frame, and likely obtain good results (except  perhaps for the fact that the dynamic part is a bit harder to reconstruct due to fine details and motion blur). Is this baseline considered here? Can the authors explain why their approach should be better than this simple baseline?
4. If the cameras are static, couldn't you just use background subtraction to quickly and easily identify the dyadic Gaussians? Do you really need to do so by training a mask?
5. The work has a strong dependency on prior works like 4D-GS and seems of consisting of relatively minor improvements and tweaks as far as I can see.

Section 3 (preliminaries) seems to be missing important equations and notation; for instance, where is the equation deriving the blending weights $\alpha_i$ computed from the corresponding Gaussians? Each Gaussian should have, in addition to position, colour and shape, an opacity factor. Where is it and how does it affect the method?

Section 4.1 should give some intuition of how the masks will be learned. The motivation for having a mask is clear, but it is not clear what is the "learning principle" that will allow an algorithm to pick up a good mask and tell which Gaussians are dynamic. It seems that the intuition is as follows: masked Gaussians are only used to reconstruct the first frame, so that the algorithm can switch them off in the reconstruction of other frames. There will be preference to switch off the Gaussian corresponding to the dynamic objects as these will not explain correctly frames other than the first. Is this intuition correct? If so, switching off Gaussian will leave large holes in the reconstruction of all frames except the first, which will also cause the reconstruction loss to increase. Wouldn't this cause problems?

In Section 5.1 we learn that the Gaussians are initialised using a "sparse point cloud". How is this obtained? Is this done via COLMAP on a frame-by-frame basis? How are the point clouds obtained from different frames reconciled in a single set of points for initialisation? Line 324 suggest that sometimes some frames are used to extract the point clouds, sometimes more frames, and line 414 suggests that point clouds for each frame cannot be obtained at all for Panpotic Sports (why?).

Section 4.2 should explain better the nature of the "relay Gaussians". The paper simply states that these replicate the dynamic Gaussian form the first stage, first frame. This replication is done for *all* the segments in which the video is broken. Does this mean that these Gaussians, which are (poorly) initialised in the first frame of the video are then used as initialisation as dynamic Gaussians for all subsequent windows, including those that are far away from the first frame? Wouldn't this make the initialisation worse and worse as segments are farther and farther away from the initial point?

Relay Gaussians are supposed to allow different video segments to communicate. However, it is not clear how this is achieved. From reading the paper, it seems that the dynamic Gaussians in different segments are, via replication, made independent. If so, how does information flow between segments?

Section 4.2 should give some intuition of why it makes sense to generate a pseudo-view by combining linearly some other views.  What is this trying to achieve? Why does it make sense to put this into the loss? What time instant is supposed to be approximated by this blending and what corresponding time parameter is put into the Gaussians for reconstructing it?

Section 4.3: (line 290) why do you need to motion the motion of background Gaussians? Aren't these supposed to be static by definition?

# Minor issues

* You should elaborate on Eq. (5) and its interpretation. Rather than saying that this idea is "inspired" by Compact3DGS, you should say that Formulation (5) is taken from Compact3DGS. Besides being a more accurate attribution, that paper at least explains the origin of Eq. (5) as a straight through estimator.
* Experiments are defined as "extensive", but in practice the method is tested on 8 scenes (6 from Panoptic Sports and 2 from VRU). I am not sure I would call this "extensive".
* It is true that improvements in PSNR are small because the dynamic part of each video is small; wouldn't this suggest to also consider PSNR restricted to the dynamic regions to better measure the improvements?

**Questions:**

Please consider the many questions in the "weaknesses" section above. In particular, I would be interested in knowing if the method assumes static cameras and whether this could simplify the detection of dynamic objects. I would also like to better understand the concept of "Relay Gaussians" and how these are supposed to move information across segments.

---

### Official Review · Reviewer_V1Xt · 2024-11-03

**Soundness:** 3
**Presentation:** 3
**Contribution:** 3
**Rating:** 5
**Confidence:** 4

**Summary:**

This paper presents a method for reconstructing highly dynamic scenes (e.g., basketball games), given multi-view data of those scenes. The paper presents a 3-stage approach: (1) figure out the frame-by-frame 3D structure of the scene, and attempt to do a static/dynamic segmentation; (2) partition the scene into 16-frame clips, and for each clip re-optimize the Gaussians, asking them to explain the full segment; (3) add hexplanes and MLPs into the optimization, to fill in the blanks. Results show that this gives better PSNR than some standard baselines when tested on very dynamic scenes.

**Strengths:**

I appreciate the PanopticSports and VRU Basketball experiments. Those are great test-beds for the techniques proposed in this work.

**Weaknesses:**

The spatio-temporal divide-and-conquer with "relay Gaussians", and also the pseudo-view rendering by mixing Gaussians from different source timesteps,  to me looks very similar to techniques used in "Dynamic Gaussian Marbles for Novel View Synthesis of Casual Monocular Videos" (SIGGRAPH Asia 2024). It would be good to cite this and maybe compare against it. That paper also argues it does better than the cited baselines on obtaining good PSNR in case with large motions, but it does so entirely with 3D Gaussians (not 4D).




"RelayGS outperforms state-of-the-arts by more than 1 dB in PSNR" -- this is not very convincing for me. PSNR is a weird metric where methods that score better do not always look better to a human. Is there another metric the authors can introduce, to verify the improvement? LPIPs may be a useful complement (though it has weaknesses of its own), and might be easy to compute. Also it may be good to separate the improvements on foreground vs background, since background can sometimes dominate the metric while the focus here is on the dynamic stuff.

**Questions:**

As far as I can tell, it is never explicitly stated in 4.1 whether this stage uses 3D or 4D Gaussians. It seems 3D & 4D Gaussians are defined in Sec 3 and then the paper doesn't say which formulation is actually used. If it is 3D Gaussians, then this sentence is confusing "The foreground objects in highly dynamic scenes often undergo significant movements across frames, making it difficult to fully capture their large-scale motion trajectory with a single set of canonical Gaussians"  -- "difficult" suggests some partial success, but if it's 3D per-frame Gaussians then no motion is captured at all. On the other hand, 3D seems plausible since it says that this stage works "without considering temporal information". These mixed messages leave me confused.

---

### Official Review · Reviewer_xMKp · 2024-11-04

**Soundness:** 2
**Presentation:** 1
**Contribution:** 2
**Rating:** 3
**Confidence:** 4

**Summary:**

The paper proposed a three-stage method for reconstructing dynamic scenes from multiple synchronised video streams. The first stage aims at learning a static-dynamic decomposition by reconstructing the first frame with a large number of Gaussians, while all other frames are reconstructed only from frames that are classified as dynamic. This classification is learned simultaneously with the reconstruction. The second stage segments the video into small clips and optimises Gaussians for each one. The final stage learns a dynamic Gaussian model to capture the dynamic parts of the scene. The method is evaluated on a public and a private dataset and shows better reconstruction quality on held-out camera views.

**Strengths:**

Long-term dynamic reconstruction is a challenging and open problem that is worthwhile to address. Gaussian splatting seems to be a good representation for this problem as it models the scene as a discrete set of points that allow for simpler parametrisation of scene dynamics than neural volumes.

A staged approach is sensible: breaking down the problem into first static reconstruction, followed by dynamic reconstruction, helps to learn a meaningful background model.

**Weaknesses:**

## Mismatch between method and evaluation
The main weakness of the paper is a mismatch between the proposed method and how it is evaluated. To my understanding (some uncertainty remains: see point “Clarity” below), the method reconstructs a dynamic video from many cameras in more or less independent segments. Each segment is initialised from the same set of Gaussians but optimised independently. This means that instead of reconstructing the video with a single set of dynamic Gaussians, no temporal correspondence is guaranteed between segments. The comparison to prior work runs those methods on the whole video. Reconstructing a whole video at once is much more difficult than 16-frame segments, which likely explains the reconstruction quality improvements. In a sense, one could even reconstruct individual static scenes for each frame, which would likely result in even better reconstruction scores.
There are two options: either one could show that the proposed method results in consistent temporal correspondences across the whole video, or one shows that prior work also performs worse on individual segments. Otherwise, the proposed method is evaluated in a more favourable setting than prior work.
This insight can also be seen in the attached videos, where the scene noticeably jumps between segments.

## Model Choices in Stage 3
Stage 3 contains several stages that seem to be ad-hoc and are not fully explained.
* Stage 1 learns a static vs. dynamic decomposition. However, Stage 3 learns a motion model for static background Gaussians. Why is that necessary?
* Position deformation scaling is unclear. The position offset is already predicted by a non-linear MLP $\Delta \mu = \phi_{\mu}(f_d)$. It is unclear what the exponential formulation brings that the MLP could not learn. Moreover, since $\gamma \in \mathbb{R}^3$ it is unclear what $e^\gamma$ represents.

## Clarity
Some parts of the paper are not self-contained or are difficult to understand:
* The learnable mask idea and formulation are taken from Compact3DGS (Lee et al., 2024). Still, the paper copies the corresponding equations (eq. 5+6) but misses the explanation that this formulation implements the straight-through estimator to allow gradients through a binary mask. Without this explanation, the formulation would seem unnecessarily complex and redundant.
* That same section then explains that this formulation is better than prior work (L249); however, the formulation itself is taken from prior work, so the improvement over said prior work is unclear.
* It is unclear what Stage 2 (Sec. 4.2) optimises. There are many variables that turn out to be constants. In the end, three frames (1,8,16) are averaged into a mean frame that is used for reconstruction. Why is this done? What problem does it solve? Is a new GS scene optimised on these average frames using the dynamic Gaussians as initialisation? Are all these Gaussians “RelayGaussians”?

**Questions:**

Beyond the questions in weaknesses above, especially the questions raised in the Clarity section, there are others:


Will the VRU Basketball Games dataset be released? It is not clear in the paper if this dataset is a contribution of the paper, or some other resource. Only a link is referenced without further explanation.

---

### Note · Authors · 2024-11-15

I have read and agree with the venue's withdrawal policy on behalf of myself and my co-authors.